# OPEN ROLE-PLAYING WITH DELTA-ENGINE

## ABSTRACT

Game roles can be reflections of personas from a parallel world. In this paper, we propose a new style of game-play to bridge self-expression and role-playing: *open role-playing games (ORPGs)*, where players are given the autonomy to craft and embody their unique characters in the game world. Our vision is that, in the real world, we are individually similar when we are born, but we grow into unique ones as a result of the strongly different choices we make afterward. Therefore, in an ORPG, we empower players with freedom to decide their own growing curves through natural language inputs, ultimately becoming unique characters. To technically do this, we propose a special engine called *Delta-Engine*. This engine is not a traditional game engine used for game development, but serves as an in-game module to provide new game-play experiences. A delta-engine consists of two components, a base engine and a neural proxy. The base engine programs the prototype of the character as well as the foundational settings of the game; the neural proxy is an LLM, which realizes the character growth by generating new code snippets on the base engine incrementally. In this paper, we self-develop a specific ORPG based on delta-engines. It is adapted from the popular animated series "Pokémon". We present our efforts in generating out-of-domain and interesting role data in the development process as well as accessing the performance of a delta-engine. While the empirical results in this work are specific, we aim for them to provide general insights for future games. [1]

## 1 INTRODUCTION

The virtual world, often more idealistic than reality, presenting a utopian escape and an alternative life, has captivated human imagination for decades. Films like "Free Guy" and "Ready Player One" have presented this vision for us. Role-playing games (RPGs) offer players the opportunity to step into a well-designed character and enjoy its growth in a virtual game world, e.g. an Egyptian guard in "Assassin's Creed", an American West bounty hunter in "Red Dead Redemption". However, conventional RPGs come with inherent limitations, where players are bound to some pre-scripted characters. As time passes, they may find themselves merely acting out someone else, without any personal connection. Rather, players desire to become another version of themselves in a parallel world. To fulfill such deepest desire for autonomy and self-expression, this paper introduces the concept of *open role-playing games (ORPGs)*, which allow players to craft their own identities, attributes, and powers, etc.

*How ORPGs are played to make each player's character distinguished*    In reality, while we are individually similar at birth, we make different choices (whether actively or passively) as we grow up afterward, which shape us into totally different people. This vision underpins the primary feature of ORPGs: players are given the autonomy to manipulate the growth of their characters and become truly unique ones. We notice that current RPGs also offer some extent of autonomy by providing players with different options when their characters grow. However, it is very hard to exhaust every possible option and fulfill the desire of every player. In this paper, we consider a generalized situation, where players are able to direct the growth with free-form natural language descriptions, e.g. "Let me learn a talent to burn the enemy". Such broad semantic space of natural language unlocks an unprecedented openness over traditional RPGs. However, the openness in ORPGs do not mean the players can become anybody or anything without constraint. All of the attempts should be contextualized within the specific game world, e.g. adhering to the physical laws and established worldview.

---

[1] Code, data, and demonstration are in our supplementary materials.

*How ORPGs are technically implemented*   As aforementioned, the character growth in ORPGs is driven by natural language, which is difficult to interpret for existing game systems. We thus propose *Delta-Engine*. It is a neural engine (Wu et al., 2024a) incorporating a large language model (LLM) (Brown et al., 2020; OpenAI, 2023; Touvron et al., 2023; Jiang et al., 2023; Team et al., 2024). LLMs can serve as a powerful non-linear function to transfer natural language instructions into some kinds of outputs, such as the engine's code. Specifically, a delta-engine is composed of two components: a base engine and a neural proxy. The base engine is the initial coding of the character. The neural proxy is an LLM, enabling the character growth by generating new code snippets that expand the base engine. A delta-engine is different from traditional game engines (e.g. Unity, UE), which serve as a platform for game development, but serves as an embedded module within the game system. It would bring an entirely novel game-play experience, where the code of a player's character is personalized and dynamically generated.

To materialize our concept, we have developed a tangible ORPG *Free Pokémon* based on delta-engines, which will be open-sourced. The characters in the game are inspired from the popular Pokémon animated series[2]. In the game, players are born with a common pokemon template. From there, they have the freedom to grow up, learning their desired talents, described through natural language, being a unique pokemon to their tastes. An ORPG is data-driven. Therefore, data acts as a significant element in the development process, for aligning the neural proxy to enhance its adaptability. However, the data needed can be very domain-specific. Given the high cost of manually annotated data, this paper explores a collaborative design approach that leverages both human expertise and LLMs to generate high-quality data.

## 2  RELATED WORK

AI-driven RPGs belong to a profound field of study that spans a series of sub-topics, e.g. role-playing (Shanahan et al., 2023; Värtinen et al., 2024; Wang et al., 2024), narrative (Wu et al., 2024c; Zhao et al., 2024a;b), world building (Bruce et al., 2024; Wang et al., 2023b), data creation (Wang et al., 2023c; Rezwana & Maher, 2023). This paper focuses on the form of the virtual world, i.e. to evolve, and proposes a special engine to enable such evolution. The basic logic of our world engine is the instruction-driven game engine (Wu et al., 2024b), a neural engine incorporating LLMs as integral components. Our engine can be triggered by a high-level evolving instruction to generate new code snippets. A relevant recent work is GameNGen (Valevski et al., 2024), a real-time game engine based on diffusion models. As opposed to GameNGen, which renders the content directly from prompts, our delta-engine generates the executable code and embed it into the base engine. The eventual rendering and operation is still done by the backbone engine, which is made by code.

The chosen playground in our work belongs to role-play games (RPGs), a genre that has seen significant advancements in recent years due to the integration of LLMs (Wu et al., 2024c; OpenAI, 2023; Touvron et al., 2023; Jiang et al., 2023; Yang et al., 2023). As opposed to these efforts, our work enhances the player experiences by offering biodiversity of virtual roles. Any role can become a truly unique one through exclusive evolution.

We are not the first to choose "Pokémon" as the topic in the research. A most recent work is Pokéllmon (Hu et al., 2024), but they do an orthogonal job from us. Pokéllmon is an LLM-based framework for powerful battle strategies based on existing pokemon characters. Free Pokémon is a novel game genre, allowing players to generate their own pokemon characters (Butler et al., 2017). In addition, our work does not focus on generating visual assets of new pokemon characters (Liapis, 2018; Geissler et al., 2020). We notice that it is also an interesting line for further developing ORPGs.

Procedural content generation (PCG) (Shaker et al., 2016; Smith et al., 2011; Summerville et al., 2018) can be another relevant line of work to ORPGs. Practically, the player's behavior won't directly affect the delta-engine, but rather go through a procedure, which eventually decides the prompt to the neural proxy. In this paper, we do not focus on the design of the evolving procedure but only study the naked delta-engine.

There is a large body of work studying AI players across various game domains, e.g. Atari (Mnih et al., 2013), Minecraft (Fan et al., 2022; Wang et al., 2023a), StarCraft, (Vinyals et al., 2019), Werewolf (Xu et al., 2023), SIMA (Team et al., 2024), CRADLE (Tan et al., 2024).

---

[2]https://www.pokemon.com/us

## 3 DELTA-ENGINE

**Base Engine**  A base engine is the initial state of the delta-engine. It depicts the prototype of the virtual world, typically a mass of objects with associated methods and basic utilities.

As any object (e.g. environment, individual) grows, it acquires new properties. As a result, its associated engine is given new code. Considering an individual born with a blank template, its initial engine may be several lines of code supporting its only walking ability. As it grows stronger and learns to run and even fly, its codebase will be updated and expanded to reflect these new properties.

**Neural Proxy**  The neural proxy is a neural wrapper around the base engine, which scales the base engine by producing new code. In our paper, it is a large language model (LLM), particularly one of those that are additionally pre-trained on code, e.g. CodeLLaMA (Rozière et al., 2023), CodeGemma (Mesnard et al., 2024).

We denote all the objects and methods of the engine at some moment as a state. The LLM proxy seeks to predict the new state moment-by-moment. To make this process efficient, we ensure that the proxy always generates the incremental code on top of the current engine state, either adding new features or overloading existing ones.

**Incremental Prediction**  Given an input and the current engine state, a delta-engine seeks to predict the incremental value. This idea can be formalized as:

$$\Delta y_t = \mathcal{F}(y_{t-1}, x_t) \tag{1}$$

where $\mathcal{F}$ is the neural proxy, $x_t$ is the input, $y_{t-1}$ is the current engine state, and $\Delta y_t$ is the incremental value of $y_{t-1}$ and $y_t$. The initial state $y_0$ is the base engine.

$y_t$ can be obtained by merging $\Delta y_t$ and $y_{t-1}$:

$$y_t = m(\Delta y_t, y_{t-1}) \tag{2}$$

where $m$ is the merge function.

**Retrieval**  $y_0$ can be super large for a complicated virtual world and $y_{t-1}$ will also become more and more as it evolves, negatively impacting the engine's scalability. We notice that evolution, both in nature and in virtual worlds, tends to occur gradually, which means each evolution step of an object is relevant to only a small fraction of the current engine. Therefore, for each prediction, we retrieve the relevant parts of the engine dynamically, denoted as $\widetilde{y}_{t-1}$, to replace the entire $y_{t-1}$ as the reference:

$$\Delta y_t = \mathcal{F}(\widetilde{y}_{t-1}, x_t). \tag{3}$$

$\widetilde{y}_{t-1}$ is a sparse version of $y_{t-1}$, which is the key to the delta-engine's scalability to very long turns.

We have the neural proxy determine the entries to retrieve itself (Yu et al., 2023). Concretely, it takes a pre-step, predicting which parts of the engine are essential for scale. To do this, we index all methods within the engine using their names and prompt the neural proxy with the skeleton overview of the engine, which only keeps the structure and method names and skips the detailed implementation. Then, we extract the implementation of the methods from the engine according to the method names.

We will illustrate a concrete case of incremental prediction in the next section.

## 4 PLAYGROUND: FREE POKÉMON

Free Pokémon is developed based on delta-engines, which allows open-ended evolution for pokemon roles. Users can write their ideas in natural language, which directly manipulates the evolution step of the roles. As a result of that, the pokemon roles are able to acquire customized abilities and moves widely different from those in official games.

Figure 1 demonstrates the Free Pokémon system. It consists of two kinds of engines, role engine and battle engine. Every single pokemon role corresponds to a role engine, which is a delta-engine. It scales at each evolution step. The battle engine is responsible to host the battles between different pokemon roles. First, we can initialize the pokemon role by providing some basic settings, e.g.

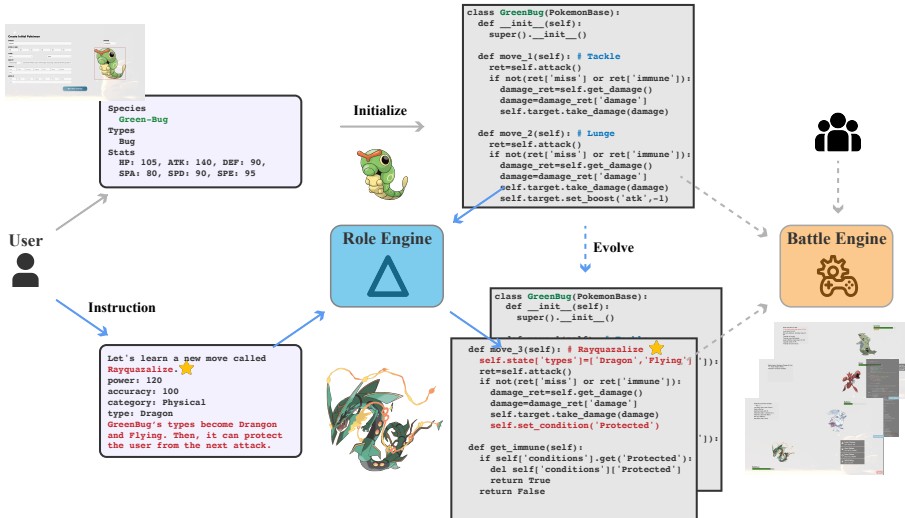

Figure 1: Free Pokémon system. Please see our supplementary materials for web demonstration.

species, types, and stats, which then will be transformed into a json format. In Free Pokémon, the initial states of all roles are almost the same, which is done by rules. Here, the user crafts a pokemon "Green-Bug" of Bug type. Then, its **role code** is initialized, with moves Tackle and Lundge. Specifically, the role is instantiated as a subclass `GreenBug` of `PokemonBase`. The `move_1`, and `move_2` correspond to its two moves respectively.

The blue stream in Figure 1 demonstrates the role's evolving process. This user provides a natural language description of his desire, letting his role learn a new move "Rayquazalize", whose secondary effect is to switch types and protect it from the next attack. This instruction will be sent to the role engine and trigger its scaling. As a result, it generates two new methods under the subclass. This scaling process will repeat for each evolution step.

Free Pokémon is an open-sourced playground for researchers interested in delta-engines. To facilitate research, the delta-engine is exposed directly to the researchers/users. They are able to manipulate the delta-engine by issuing any instructions, inputting anything they like. For example, one can craft a "Thanos" pokemon that owns super powerful moves to beat any other pokemons in one turn; even intentionally make problematic instructions to access the engine's behavior.

In Figure 2, we showcase the template we use for incremental prediction in Free Pokémon. In the first step, we prompt the neural proxy to decide the entries by providing a structural overview of the engine. Here, the proxy decides to retrieve two methods, `get_power` and `set_boost`. In the second step, we retrieve the implementation of these two methods as the reference for the neural proxy and prompt it to evolve the role. Eventually, it generates the incremental value of the engine as the response. Specifically, in the response, we develop a decorator "Increment" to merge the new code into the engine.

## 5 TRAINING DATA GENERATION

The delta-engine transfers the development process of the system to a hybrid of programming and data engineering. Developers are tasked to craft a sufficient amount of data to align the neural proxy. This process is labor-intensive. Even experienced professional designers can't always come up with fresh and innovative ideas. A recent philosophy is to synthesize pseudo data using powerful LLMs as generators (Wang et al., 2023c; Wu et al., 2024c). However, the synthetic data is somewhat low-quality and there can be significant and unknown biases within it (Shumailov et al., 2023). Instead, we adopt a human and AI co-design process, where LLMs are harnessed as assistant designers, working collaboratively with human designers. We first discuss two major demands of generated data.

```
## Input for Retrieve
You are a game programmer for Pokemon. I
will give you a structural overview of
an engine implementation of a pokemon
role and a natural language instruction.
You are tasked to retrieve the entries
you need to further implement the
instruction.

Overview:
PokemonBase
- get_power
- get_stat
- attack
- get_damage
- set_boost
- ...

Evolve instruction:
{instruction}

## Response
get_power
set_boost
```

```
## Input for Evolve
You are a game programmer for Pokemon. I will give
you a python implementation of a pokemon role and
a natural language instruction. You are tasked to
follow the instruction to evolve the pokemon role.
You can overload methods or add new ones.

Current implementation:
```python
{state}
```

Evolve instruction:
{instruction}

## Response
```python
@Increment(Pikachu,'_move_3')
def value():
  return ('Quick Attack',40,100,
  'Physical','Normal',1,['contact'])

@Increment(Pikachu)
def move_3(self): # Quick Attack
  ret=self.attack()
  if not (ret['miss'] or ret['immune']):
    damage_ret=self.get_damage()
    damage=damage_ret['damage']
    self.target.take_damage(damage)
```
```

Figure 2: Input-output template for incremental prediction in Free Pokémon. The engine is implemented using Python. For brevity, we omit some elements: the engine state $y_{t-1}$, instruction $x_t$.

**Being Novel**    Players are highly creative. For example, they won't be satisfied with similar content for long; they keep discovering novel and imaginative elements in the virtual world. Therefore, it is crucial for the delta-engine to scale to a broad range of novelty. However, we find that LLMs are not good at creating novel content based on given instances. Rather, they lean to combine existing content, such as merging two talents into a new one; or make superficial modifications, such as transforming a regular dog to a bigger dog. Such secondary data no longer enhances scalability, the emergence of which necessitates a leap into out-of-domain content.

**Being Interesting**    Interestingness further aligns the delta-engine to the player base. Our demand is to refine the data design process by picking out the interesting portion from large amount of data. As opposed to novelty, which can be straightforwardly measured using similarity scores, however, quantifying interestingness has always been a challenging task (Nelson & Mateas, 2007; Todd et al., 2024). It is highly subjective and lacks a precise definition, making it difficult to devise instructions for LLMs to assess.

## 5.1 PROTOTYPES ENHANCED IMAGINATION

We conjecture that LLMs lack or even do not have imagination; their creative outputs are still guided by the prompts they receive. However, the naive prompts e.g. "please use your imagination" fail to offer useful clues to inspire the LLM's imagination. To address this, we propose to leverage an explicit prototype, a descriptive paragraph of an entity or scene, as the imaginative foundation. It facilitates the generation of novel content by providing a concrete reference point.

Figure 3 illustrates the idea. For example, we seek to use *Tyrannosaurus* as the prototype to design a pokemon role. We retrieve the corresponding description from Wikipedia and prompt the LLM generator. The result is a novel pokemon characterized by stronger bite power, aligning with the notable feature of Tyrannosaurus. In addition to real-world entities, prototypes can also come from fictional sources. In our project, we retrieve the animals from Wikipedia, e.g. *Tyrannosaurus*, *Smilodon*, *Sperm Whale*; also retrieve the virtual creatures from "Monster Hunter"[3], a popular action video game. The distinction between the two sources is that the super-natural creatures in Monster Hunter typically lead to higher novelty of the pokemon roles designed upon them. However, these roles may go too far, increasing the likelihood of grammatical errors within the generated code, which we will discuss below.

---
[3]https://www.monsterhunter.com

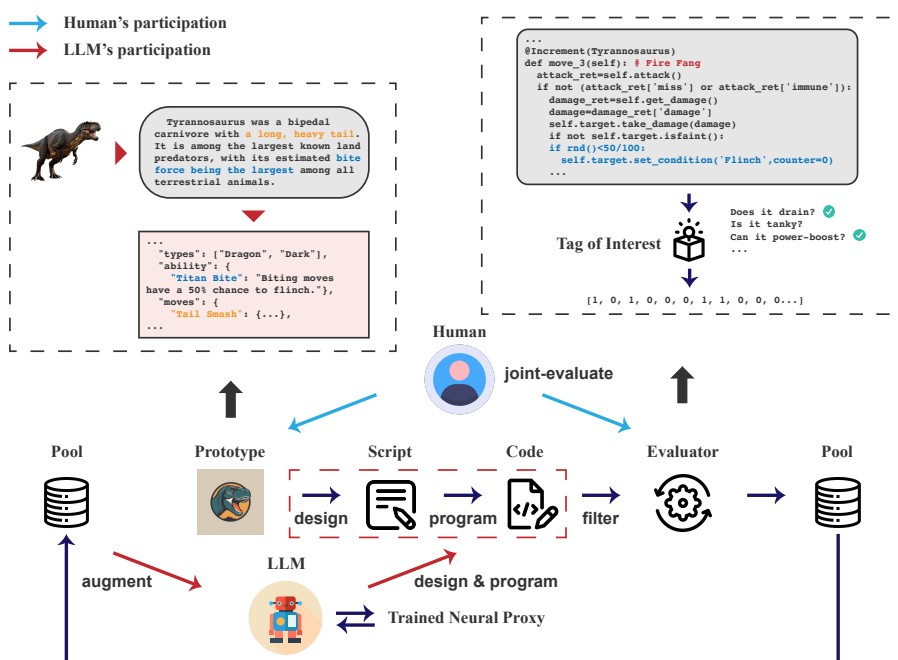

Figure 3: Human and AI design (co-design). At the top left, we illustrate the process we leverage prototypes to enhance the LLM's design. We align the descriptions of the prototype and its associated design result using colors.

## 5.2 TAGS OF INTEREST

We hypothesize that interestingness is an accumulation of potential factors that may pique the users' interest (Althöfer, 2010). This implies that the more potential factors there are, the more likely the users will find the content interesting. Therefore, we introduce an interestingness evaluator based on these factors, which we call *Tags of Interest (ToI)*. We then need a tagger to label them out given the instance.

Firstly, we establish a set of ToI. Since they vary significantly across different scenarios, this is a heuristic process. We can construct a one-dimensional "interestingness vector", where each tag is represented as one bit. For example, if a pokemon role has the ability to boost its power, the corresponding bit of this tag will be set to 1; otherwise, set to 0. We use a rule-based tagger to mine the potential tags of a role from its role code. For example, if a pokemon role can boost its power, it will inevitably overload the method get_power. Based on the interestingness vector, we set a threshold; if the magnitude of the vector does not reach the threshold, the sample will be filtered out.

## 5.3 HUMAN AND AI CO-DESIGN

Figure 3 illustrates our co-design process with the dual participation of human and AI (LLM) designers to generate the training data we need. In this process, we seek to generate two parts of data, **role script** and **role code**. The former is a natural language json script that details the pokemon role. We use a script-code pair to identify a role. Eventually, we split each role into several states as the training samples for doing incremental prediction.

Concretely, we initialize the sampling pool with 20 manually-crafted seed instances of script-code pairs. The human designer first determines the prototype and prompts the LLM designer to generate a novel role script based on the prototype. Then, the LLM designer is prompted to program the role script into the role code. More specifically, we sample 5 instances from the sampling pool to augment the LLM's coding, while only sampling 1 instance to augment its designing. A key observation is that the in-context instances of other role scripts may bias the effect of the provided prototype, incurring low creativity of the response. On the other hand, in-context instances act as useful references for the

LLM to generate high-quality role code since the programming step does not rely on creativity but accuracy. The LLM designer we use is either of GPT4 or Claude3. The newly designed script-code pair will be sent to the evaluator, which is a joint process with both rule-based and manual strategies. First, code that fails to compile or introduces new methods yet without calling them will be filtered out. Second, code that fails to pass the interestingness threshold will also be discarded. After the rule-based filtering, the human designer makes the final check on the script and code. Eventually, we place the new instance into the sampling pool ready for the next cycle of design.

An important trick is that, we replace the third-party LLM designers (GPT4/Claude3) with one of the trained neural proxy in the middle of the design process. We find them, yet powerful, still struggle with the nuanced requirements of the programming problem in our project, providing low-accuracy responses, while the trained model can tackle much better.

The incorporation of AI greatly accelerates the creative process of human designers (Rezwana & Maher, 2023), creating high-quality data. In Figure 3, we observe that human designers mainly act as a prototype designer and a joint evaluator to refine the eventual instances.

## 6 EXPERIMENT

This section reports our experiments. The results are based on our chosen domain Free Pokémon. Nonetheless, if the proposed methods effectively work on our domain, it is highly promising to generalize them in the future.

### 6.1 BASIC SETTING

To access the quality of the co-designed data, we prepare another set of data of the same size purely synthesized by Claude3 . Specifically, the synthetic data is generated using a similar pipeline as in Figure 3. The difference is that we automatically sample the official pokemon roles as the prototypes, while canceling the manual evaluation step, since these two steps necessitate human participation. We show the data statistics in the upper half of Table 1.

On the other hand, we prepare two sets of test data, corresponding to **easy** and **hard**. The easy-level data comprises 19 existing pokemon roles, all of which have appeared in official pokemon games. We sample them from the internet. This set of data is easier because the majority of roles in the training data, including purely synthesized and co-designed, inevitably share similarity with the existing ones. Their distributions are closer as a result. To deeply access the scalability of the engine, in addition, we invite 10 volunteers to manually craft the hard-level data. All of them are not only experienced in playing pokemon games, but also have a wealth of experience with a wide range of games, greatly allowing them to design novel pokemon roles. Eventually, we obtain 16 original role scripts. We manually program them and obtain the ground truth role code. Beyond originality, volunteers are asked to craft more moves and abilities for one role, which helps us to better evaluate the scalability. From Table 1, we observe that the number of evolution steps and sentence length of hard-level data is much more than those of easy-level data.

We fine-tune the CodeGemma-7b model (Mesnard et al., 2024)[4]. CodeGemma is a code LLM that is additionally pre-trained on a large number of code corpora. We train each model using LoRA (Hu et al., 2022) with $r = 8$, $\alpha = 32$, learning rate 1.5e-4, and batch size 4 for 5 epochs.

We report two scores.

*Exe%*: We calculate the success rate of executing the role code on top of the engine. Specifically, we randomly synthesize 100 roles as imaginary opponents and have the role under test to battle with them, choosing a random action each time. One success will be counted if all actions are executed successfully against all opponents.

*Acc%*: We further verify the accuracy/correctness of the role code. This step is done by GPT4, which is prompted to compare two code snippets. Note that we calculate the correctness only among the successfully executed code.

---

[4] https://huggingface.co/google/codegemma-7b-it

Table 1: Upper: Dataset statistics, in order: the number of roles we created, the number of samples, the average number of evolution steps of each role, and the average sentence length. To calculate the sentence length, we use the CodeGemma tokenizer. Lower: Results on different test sets. ✓ indicates the 100% score. "Retr." refers to the retrieval technique.

| *Statistic* | **Roles** | **Samples** | **#Evolves** | **#Length** |
|---|---|---|---|---|
| Sy. Train | 167 | 500 | 3.0 | 1197.8 |
| Co. Train | 175 | 502 | 2.9 | 1167.3 |
| Easy Test | 19 | 43 | 2.3 | 997.2 |
| Hard Test | 16 | 87 | **5.4** | **1841.6** |

| | **Easy** | | **Hard** | |
|---|---|---|---|---|
| *Performance* | Exe | Acc | Exe | Acc |
| CodeGemma w. Sy. | 95.3 | 86.0 | 86.2 | 58.6 |
| CodeGemma w. Co. | ✓ | 95.3 | 90.8 | 83.9 |
| CodeGemma w. Co. w. Retr. | ✓ | ✓ | **92.0** | **89.7** |

## 6.2 Main Results

From the lower half of Table 1, we find that the two models trained on synthetic data and co-designed data (Sy. & Co.) perform comparatively on the easy test set. This is due to the closer gap between training and test data in this scenario. More specifically, the co-designed data by both humans and AI performs slightly better. The resultant model and its retrieval-augmented version achieves a full Exe rate, and the latter also achieves a full Acc rate.

On the other hand, the hard test data delivers a large distribution gap from the training data, leading to noticeable performance drop across all three model counterparts. More importantly, the gap between Exe and Acc becomes more pronounced. The elevated Exe rate indicates that the trained model is inclined to respond with executable code even if the input role is unfamiliar. However, the accuracy of code is much harder to fulfill. We find that the model trained only on synthetic data is notably weaker. This is due to the fact that the synthetic data is too limited and doesn't provide useful signals for out-of-domain generalization. In contrast, the co-design process produces high-quality data with out-of-domain signals, which significantly generalizes the model, improving the Acc rate from 58.6 to 83.9. Furthermore, we find that the retrieval technique also shows its positive impact, further improving Acc to 89.7.

In addition to out-of-domain generalizability, the delta-engine's scalability includes its scaling performance through long evolution steps. To do this, we conduct another experiment where we continuously scale the delta-engine. Specifically, we randomly sample abilities and moves from the existing database and repeatedly prompt the neural proxy to scale, until it gives a non-executable response. The result will be a "super patchwork" pokemon role. We repeat this process 100 times. Figure 4 shows two histograms, where we demonstrate the scaling performance of the engine through the evolution steps and the engine size. The engine size refers to the number of tokens of its current code. Intuitively, we observe that as the evolution increments, the performance exhibits a pronounced degradation. More specifically, as the evolution accumulates to 20 steps, only half of the cases give an executable response. A similar trend can be seen as the engine size accumulates to 5000. Note that, the length limit of the CodeGemma model is 8192. However, we find that the introduction of the retrieval technique brings a nice scalability in the face length increasing. The resultant model maintains a nice performance up to 30 evolution steps. This is because the retrieved engine state is much smaller compared to the entire engine. We illustrate a case on the right side of Figure 4. The model only retrieves the `type_change` method from the engine as the context for the following incremental prediction.

So far, we haven't explore larger-sized LLMs, though they can be promisingly stronger.

## 6.3 Data Analysis

To further investigate why co-designed data outperforms synthetic data and the distinction between easy and hard test data, we take a closer look into the underlying data distribution. Therefore, we visualize the data points of pokemon roles from two distinct views in Figure 5. Specifically,

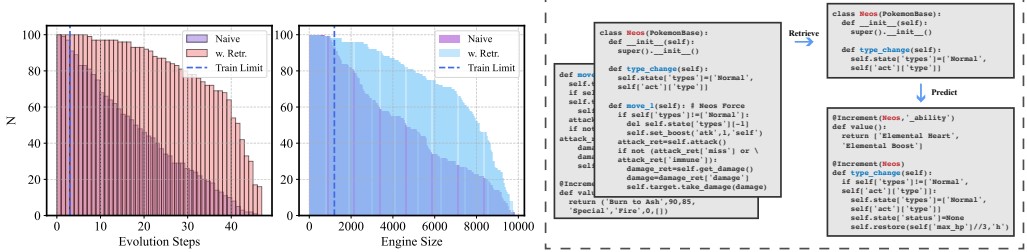

Figure 4: Histograms of 100 sampling. We highlight the number of evolution steps in the training data as a baseline. On the right, we show a concrete case of the retrieval process.

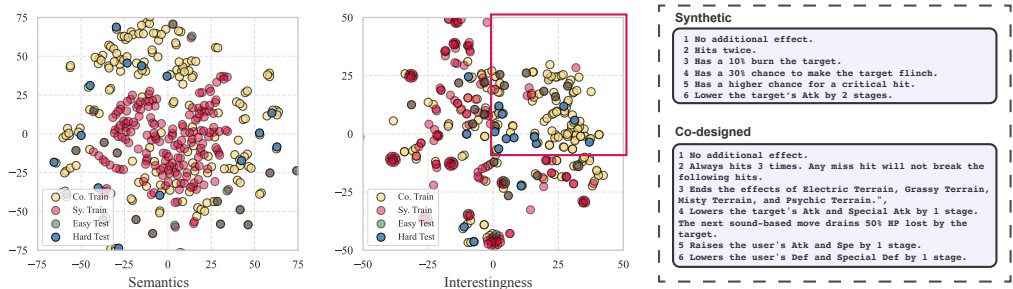

Figure 5: Comparison of the roles crafted by different methods and a concrete case on the right. We visualize them from the semantics and interestingness space.

we apply the sentence embedding model[5] to encode the role descriptions into vectors and obtain the interestingness vectors based on ToI. Then, we apply t-SNE to project all vectors into a two-dimensional space.

From the left side of Figure 5, we observe that the co-designed data points nearly encompass all synthetic data points, with the distribution of the latter exhibiting more converged. It highlights the fact that the co-designed outweighs the synthetic one in terms of semantic diversity, thus enhancing the training. We show a case on the right side. They belong to the same pokemon role yet are generated by different methods. We find that the synthetic data easily falls into an identical pattern, while the co-designed one exhibits great diversity. We find a quite different vision when we segment the data from interestingness. A similar observation is apparent that the co-designed data points continue to cover almost all synthetic ones. In particular, we notice that in the upper right, the area we have highlighted with a red box, there is a blind spot of the synthetic data. It means that the model trained solely on synthetic data, fails to capture meaningful signals from the test data in this area. Furthermore, we observe that most of the hard test data points, which are crafted by humans, are distributed in this area.

## 7 CONCLUSION

This paper concentrates on the evolving nature of the virtual world. We model this by proposing the delta-engine. The experiments are made on our self-developed playground Free Pokémon. This work is the initial attempt into the study, opening up a wealth of valuable topics for future research, e.g. developing a fully realized virtual world system, studying better training techniques to align the neural proxy, addressing the safety concerns.

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
