# OpenReview forum: "Evolving Virtual World with Delta-Engine"
_ICLR.cc/2025/Conference — Submitted to ICLR 2025_

### Official Review · Reviewer_FHDi · 2024-11-02

**Soundness:** 1
**Presentation:** 1
**Contribution:** 1
**Rating:** 1
**Confidence:** 4

**Summary:**

This manuscript is not suitable for publication. I cannot understand it. It’s full of strange, nonsensical sentences and descriptions. It uses all sorts of unusual jargon without ever defining it. Sometimes it feels like it is about to veer toward making sense, only to confuse me again. I am unable to judge the quality or purpose of any experiments done (or their motivations) because the writing is so confusing. I am not sure if it was written by a poor language model (the best ones would write something more understandable), or scientists that have not undergone any or at least proper training, but it is not ready for publication. If this is a well-intentioned attempt, I apologize for the harsh review, but I recommend you work closely with trained scientists to learn how to write a clear manuscript that meets the bar for scientific manuscripts. There are some hints of some good ideas, but they are far from being dealt with properly enough to evaluate them, let alone endorse the entire manuscript for publication.

This is the shortest review I have ever written in over 20 years in the field. It is hard to think of what else to say except the manuscript is unintelligible enough that there’s not much more to say than that.

**Strengths:**

See main review.

**Weaknesses:**

See main review.

**Questions:**

I do no understand almost anything, so my question/challenge is to try to explain everything clearly in your next submission of the work.

---

> ### Comment · Reviewer_FHDi · 2024-11-13
> **Confusing Writing**
>
> I would also point out that all of the reviewers also mention that the writing left them confused, and in most cases to serious degrees. Frankly, I am surprised how high some of the scores are given that. Thus, in my view, there is a consensus on this issue.

---

### Official Review · Reviewer_judw · 2024-11-03

**Soundness:** 1
**Presentation:** 1
**Contribution:** 1
**Rating:** 1
**Confidence:** 4

**Summary:**

In this paper the authors present a concept for a continually expanding game engine which they call a "delta engine". The authors pitch is that an LLM can be used as a wrapper to determine how best to update an engine written in code based on some instruction or signal. They then present a toy environment based on Pokemon where they use a fine-tuned LLM and prompts to create new Pokemon or add additional moves to a Pokemon. They then present an experimental evaluation in which they demonstrate their approach can accurately recreate existing Pokemon and Pokemon designed by human volunteers.

**Strengths:**

The core concept of the delta engine is an interesting one, though it's unclear whether this is substantially different than any game engine that includes generated content from a code generator LLM. Generating new Pokemon moves or base stats is technically novel, but it is not surprising that LLMs can generate code that works for these purposes.

**Weaknesses:**

This paper has a very large number of issues in its current state. To organize my feedback I'll introduce these issues in each section.

### Introduction

The introduction has the primary issue of not being particularly relevant to the actual research work that has been done. There is no clear pathway from the authors prompt-based Pokemon generation to the virtual worlds they describe. It would be better to focus the paper on the specific research project being presented. I would further suggest that the authors remove all unsupported claims like "Its codebase will become more and more along with the world’s evolution" or "Delta-engines can serve as the basic components of the world to simulate their evolving processes, encompassing roles, surroundings, props, and other integral components", given there's no proof of either in this paper or in prior work. Similarly, I would suggest the authors remove tangential sentences like "The evolution is triggered by specific signals within the world, e.g. observations, behavior, and events.". The introduction also introduces two other recurring issues in the paper. The first is that many of the figures are not legible, like the choice and size of font in figure 1. The second is that the language is very poor with many grammar or inappropriate wording issues, such as "Such dynamics is unpredictable"->"Such dynamics are unpredictable" or the use of "Pokemon role" when I believe the authors may have meant "Pokemon character".

The authors notably claim that code, data, and a demonstration are available in the supplementary materials, but this is untrue. There is a small Pokemon battling clone with some pre-generated elements.

### Related Work

The authors cite a great deal of prior work, which is great. However, many of the citations are not relevant to this work. For example, the authors do not need to list all the AI work across various games in the final paragraph. Instead, I would have recommended that the authors discuss prior work in generating code for game characters [1,2]. In addition, the authors likely should have touched on prior work on generating Pokemon, though the majority of this prior work is focused on generating Pokemon-like visuals [3,4,5], some of the prior work does touch on descriptions [3] or type information [5].

It is also notable that GameNGen is not based on prompts.

### Delta Engine and Playground: Free Pokemon and Training Data Generation

I am grouping all three of these as they make up the system overview equivalent sections of the paper. In general, the authors would ideally have included all technical aspects of the work in sufficient detail that they could be replicated. But this is not the case, the authors do not include their representation of Pokemon or rules (except via examples), they do not give their prompts or prompt structure, and most importantly they do not give how their co-creative setup works. Part of the problem here may be the lack of clarity (writing and language issues) in the paper.

### Experiments

The authors present an experiment to show that their approach can recreate existing Pokemon and that it can recreate Pokemon created by human "volunteers". The issues here are primarily with the volunteers with the experiments otherwise being very reasonable for evaluating Pokemon generation. The authors do not specify if they had ethics approval for this human subject work or if the participants were compensated. It's also unclear what prior knowledge they had or what their relationship is to the authors. Without full methodological information it is impossible as a reader to judge the validity of their date, making the results related to it similarly difficult to trust. Similarly, it's unclear what the co-creative experience was or who the humans were who took part in it, making it very unclear how to interpret the "& CO." results. These issues remove any generalizable knowledge other researchers may have been able to take from this experiment.

Figures 5 and 6 similarly have issues in terms of how they were made and what they mean. For Figure 5, it's unclear if this experiment was run a single time or what all the colours and lines indicate. For Figure 6, it's unclear how the authors created their semantics and interestingness spaces or how they are projected into two dimensions.


1. Butler, Eric, Kristin Siu, and Alexander Zook. "Program synthesis as a generative method." Proceedings of the 12th International Conference on the Foundations of Digital Games. 2017.
2. Sorochan, Kynan, and Matthew Guzdial. "Generating real-time strategy game units using search-based procedural content generation and monte carlo tree search." arXiv preprint arXiv:2212.03387 (2022).
3. Geissler, Dominique, et al. "Pokérator-unveil your inner Pokémon." 11th International Conference on Computational Creativity, ICCC 2020. 2020.
4. Liapis, Antonios. "Recomposing the pokémon color palette." Applications of Evolutionary Computation: 21st International Conference, EvoApplications 2018, Parma, Italy, April 4-6, 2018, Proceedings 21. Springer International Publishing, 2018.
5. Gonzalez, Adrian, Matthew Guzdial, and Felix Ramos. "Generating gameplay-relevant art assets with transfer learning." arXiv preprint arXiv:2010.01681 (2020).

**Questions:**

1. Have I substantially misunderstood the authors work?
2. What was the methodology for the human subject study/volunteers?

**Details Of Ethics Concerns:**

There is a potential need for an ethics review in terms of the research practice here. The authors mention volunteers but there's no clear human subject study or ethics approval. This makes it unclear the ethics of the volunteer's participation or whether there was some pressure to participate. It's also unclear if the volunteers include the authors, which would be a clear issue in terms of biasing the results.

---

> ### Author Response · Authors · 2024-11-13
>
> Thank you for your review. Some of the points are really useful and we appreciate that.
>
> However, we find that most of the review does not focus on the technical novelty and technical contribution of the work, which is a pity. Instead, your rating is strongly based on the clarity of our paper and implementation details. Indeed, these parts can be quickly addressed or improved in the revision. Based on your review, we make some clarifications below.
>
> 1. **Implementation details and replication**
>
> We open-source the entire Free Pokemon project on the Internet and anyone can download and play, even check and modify our code, use our trained CodeGemma model. Is this not enough for replication?
>
> For the details you mentioned in the review, e.g. the game rules of Pokemon games. We believe they are minor for our paper and there is no space left in the submission.
>
> 2. **Some details that we have already written in the paper**
>
> Respectfully, you have missed some of the details we have written in the paper. You do not have enough time to read the paper and we understand that. Thus, we further clarify them.
>
> * Figure 5: we mention that we repeat this process for 100 times in line 420.
>
> * Figure 6: we have clearly mentioned the method that we use to project the semantics and interestingness vectors in line 463 to line 465. Please read them if you like.
>
> 3. **Citation recommendation**
>
> Thank you for the recommendation. We find [1] and [3] helpful, while the others, not highly related to our work. For example, we do not focus on generating visual assets in our work. Sure, we would like to mention them in our paper.
>
> 4. **Unfaithful rating**
>
> We appreciate that some of the points in the review are useful. The rating is still unfaithful. It seems that you get the general idea of our work and find our work technically novel though not surprising to you as you mentioned. However, **we have found no indication that there are technical flaws in our work or the entire contribution of our work is trivial**. So why all of our paper is rated with 1 score?
>
> To conclude, we thank your time to review our paper. However, we are not convinced of your rating. In addition, we are confident that our work is solid and could make great contribution to the concerning community.

---

> ### Author Response · Authors · 2024-11-13
>
> Keeping up with our last comment
>
> 5. **Prompts**
>
> We have already demonstrated our prompts for inference in Figure 3. For the prompts we use to design new Pokemon roles, we have also open-sourced them in the supplementary materials.
>
> We show them below:
>
> ```json
> You are professional designer for Pokemon games. I will give you a paragraph describing a specific animal. Use this animal as the prototype to creat a new Pokemon.
> IMPORTANT:
> 1. You should craft entirely new abilities and new moves.
> 2. You should return 1~2 abilities and 3~4 moves.
> 3. The types of moves should be broad.
>
> Your reference animal:
> Tyrannosaurus was a bipedal carnivore with a massive skull balanced by a long, heavy tail. It is among the largest known land predators, with its estimated bite force being the largest among all terrestrial animals. By far the largest carnivore in its environment, Tyrannosaurus rex was most likely an apex predator, preying upon hadrosaurs, juvenile armored herbivores like ceratopsians and ankylosaurs, and possibly sauropods.
>
> You should return like the following format. Do not return any additional words.
> Output example:
> {
>   "species": "Arceus",
>   "types": [
>     "Normal"
>   ],
>   "gender": "Neutral",
>   "ability": {
>     "Wonder Guard": "This Pokemon can only be damaged by supereffective moves and indirect damage.",
>     "Corrosion": "This Pokemon can poison or badly poison a Pokemon regardless of its types."
>   },
>   "moves": {
>     "Thousand Arrows": {
>       "power": 90,
>       "accuracy": 100,
>       "category": "Physical",
>       "type": "Ground",
>       "effect": "This move can hit airborne Pokemon, which includes Flying-type Pokemon."
>     },
>     "Substitute": {
>       "power": 0,
>       "accuracy": 100000,
>       "category": "Status",
>       "priority": 0,
>       "type": "Normal",
>       "effect": "The user loses 1/4 of its maximum HP and sets a substitute with the same amount of HP to take damage from attacks for it. The substitute is removed once enough damage (1/4 of max HP) is inflicted. Fails if the user does not have enough HP remaining, or if it already has a substitute."
>     },
>     "Toxic": {
>       "power": 0,
>       "accuracy": 90,
>       "category": "Status",
>       "type": "Poison",
>       "effect": "Badly poisons the target."
>     },
>     "Protect": {
>       "power": 0,
>       "accuracy": 100000,
>       "category": "Status",
>       "priority": 4,
>       "type": "Normal",
>       "effect": "The user is protected from attacks made by other Pokemon during this turn. Fails if the user uses this move last turn."
>     }
>   }
> }
> Your output:
> ```
>
> Here, the in-context sample is randomly sampled, as we mentioned in our method.
>
> 6. **Volunteers in our experiments**
>
> **We have mentioned the features of the 10 volunteers in line 362 to line 364.** "All of them are not only experienced in playing pokemon games, but also have a wealth of experience with a wide range of games, greatly allowing them to design novel pokemon roles."
>
> Why should we mention the relationships between us and volunteers? This is personal. Does this create a barrier to understanding our evaluation?
>
> First, in Figure 6, **we have showed the strong distribution gap in the semantics and interestingness space between existing Pokemon (east test set) and human-crafted Pokemon (hard test set)**. If this is not convinced to you, **please make a clear question why this is not**. If you just miss this part, please read them if you like.

---

> ### Author Response · Authors · 2024-11-13
>
> We will show two cases, one for existing Pokemon and the other for volunteer-designed Pokemon. Note that, we open-source all the Pokemon roles assets in our project. Anyone can check them to figure out.
>
> Existing Pokemon:
> ```json
> {
>   "species": "Blastoise",
>   "types": [
>     "Water"
>   ],
>   "gender": "Male",
>   "ability": {
>     "Torrent": "At 1/3 or less of its max HP, this Pokemon's offensive stat is 1.5x with Water attacks."
>   },
>   "moves": {
>     "Shell Smash": {
>       "power": 0,
>       "accuracy": 100000,
>       "category": "Status",
>       "type": "Normal",
>       "effect": "Lowers the user's Defense and Special Defense by 1 stage. Raises the user's Attack, Special Attack, and Speed by 2 stages."
>     },
>     "Hydro Pump": {
>       "power": 110,
>       "accuracy": 80,
>       "category": "Special",
>       "type": "Water",
>       "effect": "No additional effect."
>     },
>     "Ice Beam": {
>       "power": 90,
>       "accuracy": 100,
>       "category": "Special",
>       "type": "Ice",
>       "effect": "Has a 10% chance to freeze the target."
>     },
>     "Aura Sphere": {
>       "power": 80,
>       "accuracy": 100000,
>       "category": "Special",
>       "type": "Fighting",
>       "effect": "This move does not check accuracy."
>     }
>   }
> }
> ```
> Volunteer-designed Pokemon:
> ```json
> {
>   "species": "Graphal",
>   "types": [
>     "Dark",
>     "Ghost"
>   ],
>   "gender": "Male",
>   "ability": {
>     "Dark World": "On switch-in, summons Dark World for 5 turns. During this time, Dark-type attacks cause 1.5x damage.",
>     "Lord of Dark": "For one chance, when this Pokemon's HP zeroes, it revives with 50% maximum HP."
>   },
>   "moves": {
>     "Dark Rainbow": {
>       "power": 100,
>       "accuracy": 100,
>       "category": "Special",
>       "type": "Dark",
>       "effect": "Has a 30% chance to flinch the target. In Dark World, this move's type effect is always 1."
>     },
>     "Flash Cannon": {
>       "power": 80,
>       "accuracy": 100,
>       "category": "Special",
>       "type": "Steel",
>       "effect": "Has a 10% chance to lower the target's Special Defense by 1 stage."
>     },
>     "Dark Dealings": {
>       "power": 0,
>       "accuracy": 100000,
>       "category": "Status",
>       "type": "Dark",
>       "effect": "Raise the user's Atk., SpA. and Spe. by 2 stages at the cost of 50% of maximum HP."
>     },
>     "Dark World": {
>       "power": 0,
>       "accuracy": 100000,
>       "category": "Status",
>       "type": "Dark",
>       "effect": "Summons Dark World for 5 turns."
>     },
>     "Dark Budokai": {
>       "power": 0,
>       "accuracy": 100000,
>       "category": "Status",
>       "type": "Dark",
>       "effect": "For 3 turns, once the opponents use a status move, at the end of the turn, they loses 1/3 max HP and the user restores 1/3 max HP for that."
>     },
>     "Oblivion Wing": {
>       "power": 80,
>       "accuracy": 100,
>       "category": "Special",
>       "type": "Flying",
>       "effect": "The user recovers 3/4 the HP lost by the target."
>     }
>   }
> }
> ```
>
> We see that the **human-crafted Pokemon has totally different ability and move effects (e.g. reviving, manipulating type effects), compared to the existing one**, which is much harder. This point, is clearly mentioned in our paper, line 365 to line 367.

---

> > ### Comment · Reviewer_judw · 2024-11-13
> > **Re: Official comment by authors**
> >
> > Thanks to the authors for their quick and lengthy response. I will attempt to summarize my response to their responses:
> >
> > 1. Implementation details and replication: From the text in the paper what is provided in the supplementary material is a "demonstration". The supplementary material code seems to include examples of generated Pokemon, but not the code to run all of the experiments in the paper. If I have misunderstood, I would ask the authors to point to the parts of the supplementary material that include the code for the experiments in the paper.
> > 2. Some details that we have already written in the paper: I think I may have not expressed these concerns clearly, apologies. I did read line 420, I recognize this was repeated 100 times. My concern was not the number of iterations but how many seeds were used. I also did read lines 463-465. The text states the authors "obtain the interestingness vectors based on ToI" but the detail on ToI is insufficient in the paper, with the authors mentioning a "rules-based tagger" and an example. I would also ask the authors to please come to this discussion in good faith. I understand the authors disagree with my review and I am amenable to discuss my concerns, but remarks that play down the time spent on my review are not appreciated.
> > 3. Citation recommendation: Sure, that's fair
> > 4. Unfaithful rating: As stated in my response to (2) I would encourage the authors to take to this discussion with a more professional attitude and to avoid attacks on a reviewer's ability or character
> > 5. Thanks to the authors for pulling out the prompt, that's helpful to see. However, reviewers are not required to look in the supplementary materials, and so it would be beneficial to include prompt structure in the paper/appendix.
> > 6. There is a clear risk for bias if the volunteers felt pressured in some way to take part in the human subject study. For example, if one of the authors is an instructor, employer, friends, or family of the volunteers. I am not saying this is the case but this is why these sorts of details are necessary to divulge. This is typically why ethics approval is sought for human subject studies, as this ensures that there was no undue bias on volunteers to take part.

---

> > > ### Author Response · Authors · 2024-11-14
> > >
> > > Thank you for your further feedback. Please allow us to point out that **most of your criticism is due to an unsubtle reading of our work.** For example, **the prompts we use to generate the training data are already included in the supplementary.** Please see the `prompt` dir.
> > >
> > > 1. **Implementation details and replication**
> > >
> > > **There is the demonstration in our supplementary material**. Please refer to `index.html`. It is the demo that we developed using html5 and typescript. ""Please run it."" While we have already written the notes in the README, we can still list them here.
> > >
> > > First, run the `app.py` file to start the backend.
> > >
> > > Second, open the `index.html` file on your localhost.
> > >
> > > You seem to misunderstand the meaning of a "demonstration" and experimental code. Of course, **our project can be easily replicated by anyone.**
> > >
> > > Here is the CodeGemma model we have trained. https://drive.google.com/file/d/1iVvZgBHK71yMJGzrcz4LKLSN_Y-vt4Ir/view?usp=share_link
> > >
> > > Here is the code we generate the training data from the assets provided in the supplementary. https://drive.google.com/file/d/1xJLMgnLQlgs-7zW2Lud7_Mo4hzUFRoNr/view
> > >
> > > If you query for further code to replicate our experiments, please feedback.
> > >
> > > Importantly, we want to clarify that, **every code of our project can be entirely found on the Internet**, ready for people's check. Of course, you can choose not to see them, which is up to you.

---

> > > ### Author Response · Authors · 2024-11-14
> > >
> > > We appreciate your feedback and the time you took to make the review. Do our clarifications address your confusion?
> > >
> > > Additionally, we kindly request that you pay more attention to the **overall technical aspects of our work** (e.g. soundness, originality, and validity of our methods), rather than clarity and the minor details you probably missed, which is the job that an ICLR reviewer is supposed to do. **And, what exactly was it that made you give us such a low rating?**

---

> ### Author Response · Authors · 2024-11-14
>
> 2. **Some details that we have already written in the paper**
>
> Respectfully, we are afraid you do not understand our methods and experiments at first.
>
> We further clarify our experiment for Figure 5. In line 419 to line 420, we mentioned that "we randomly sample abilities and moves from the existing database and repeatedly prompt the neural proxy to scale, until it gives a non-executable response." This sampling process, we repeat for 100 times and **each sampling is independent**, which means all of them have a different seed. The purpose is to reduce variance.
>
> For Figure 6,  **your original question in your review is** "it's unclear how the authors created their semantics and interestingness spaces or how they are projected into two dimensions.". These parts are already included in the paper. Do I misunderstand your question?
>
> **Sure, we can further answer your second question**. The rule-based tagger is use to construct the interestingness vector in ToI. We have written an example in line 312 to line 313, "For example, if a pokemon role can boost its power, it will inevitably overload the method get power.". You may still not get the idea. That's ok. Here is the example code here.
>
> ```python
> tag_dims = ['drain', 'heal', 'recoil', 'stat-boost', 'weaken', 'status', 'condition', 'env', 'priority', 'burn', 'paralysis', 'sleep', 'poison', 'freeze', 'power-boost', 'damage-boost', 'protect', 'shield', 'type-change', 'accuracy', 'crit', 'substitute', 'flinch', 'confusion', 'multi-hit', 'stat', 'type-effect']
>
> def is_power-boost(code):
>     global vector
>     if "def get_power" in code:
>         vector["power-boost"] = 1
> ```
>
> We acknowledge that these parts of the code is not attached in the supplementary. If you want, we can send you. However, please let us know **how these parts make it hard for you to verify the contribution of work** and **why this is so important to you to make a low rating for our paper**.
>
> 3. **Volunteer issues**
>
> We appreciate your further clarification of your concern. We are willing to discuss more on this. All 10 volunteers are my friends and they are college students. As I mentioned in the paper, all of them are not only experienced in playing Pokemon games, but also have a wealth of experience with a wide range of games. **This greatly allows them to design novel Pokemon roles.**
>
> On the other hand, the volunteers do not feel pressure during their contribution. They are all Pokemon lovers. Before engaging our work, they already had the prototype ideas for their uniquely designed roles. Thus, they love their jobs and finish them with ease. Of course, I instructed them before starting to ensure that the newly-designed roles are strongly divergent from existing ones. However, this requirement is easy for them to fulfill.
>
> Back to our experiments, which plays a more important role in our paper. Our main focus is to **ensure the distributional gap between easy test set and hard test set.** **This allows us to evaluate the performance (e.g. generalize to OOD)** of the delta-engine in the experiments. Figure 6 and the aforementioned cases also prove this point. Do you agree with us on this point? If not, please raise and we will answer for that.
>
> However, if your focus is about ethic, e.g. volunteer's occupation, mood, which we believe are **irrelevant to the soundness of our experiments**, we are sorry that they cannot be addressed.

---

### Official Review · Reviewer_jL87 · 2024-11-03

**Soundness:** 2
**Presentation:** 2
**Contribution:** 1
**Rating:** 3
**Confidence:** 2

**Summary:**

The paper introduces a novel virtual world engine called Delta-Engine, that evolves dynamically based on user behavior. Delta-Engine is composed of a base engine and a neural proxy. The neural proxy generates new content on the base engine through incremental prediction. The technical implementation uses retrieval techniques to enhance the connection between the neural proxy and the base engine. Additionally, it introduces a human-LLM collaborative design to ensure that the generated content is novel and engaging.

To be honest, the reviewer fails to identify what this work actually does / the contributions. The above summary was written based on the paper text. The reviewer is happy to revise the review if the following comments/questions can be addressed.

**Strengths:**

-  Appendix provides a demo.

**Weaknesses:**

- The manuscript lacks clarity, making it difficult for readers to understand the significance and motivation behind the study.

- The absence of a clear problem definition undermines the overall coherence of the paper. The concept of the Delta-engine is vaguely defined, and the paper does not provide definitions for fundamental terms, such as “engine state,” “new features,” and their relationships with virtual environmental dynamics generated by the neural proxy.

- Additionally, the description of the engine and evolving world lacks sufficient details, making it challenging to evaluate the feasibility and innovation of the proposed approach. The authors mention that the engine must address challenges in both algorithm design and data management. However, the experiments on AI co-design and the synthetic data generation process remain unclear in terms of how they work.

**Questions:**

Please refer to the comments. Can the authors clarify the contribution of the work?

---

> ### Author Response · Authors · 2024-11-14
>
> Thank you for your review. We understand that you do not fully understand our paper. We should make our paper easily to understand for a broader audience.
>
> As you require, we clarify the contribution of our work. Our contribution is multi-aspect.
>
> 1. **Evolving world**
>
> Why every living person looks greatly different? Our answer is that people can grow up as they wish, e.g. learn different skills. This is the idea of the evolving world.
>
> 2. **A new engine proposed to facilitate the evolving world**
>
> For example, as apart from existing RPGs, the roles in the game can be greatly similar if the player has finished the most of the game content. They can not learn new skills and meet new NPCs. To address this, delta-engine generates new code according the users' behavior to let the virtual world keep evolving, providing new game experience for players. For example, the player's role can learn his personal skills by the delta-engine.
>
> 3. **Training data generation**
>
> The purpose of generating training data is to enhance the LLM ability within the delta-engine. Our focus is the novelty and interestingness of the data, which is aligned with real player distribution. Thus, we propose two techniques, prototypes enhanced imagination and tag of interest (ToI).
>
> 4. **Potential useful role data**
>
> We construct a large amount of data of Pokemon roles in our project. All of them are in high quality. If you like, you can refer to them from the supplementary material in `asset`. They are also potentially useful for future researchers.
>
> Eventually, we are confident that our work can make great contribution to the community. If you are confused for some parts in our paper, please let us know. We are willing to clarify. Thank you.

---

> ### Comment · Reviewer_jL87 · 2024-11-16
> **Response to Official Comment by Authors**
>
> Thanks for the responses.
>
> I was following this paper since the first response posted by the authors, to figure out what have been done in this work. After looking at other reviewers' comments and authors' responses to all reviewers, I figured out how to explain why I failed to understand the paper and why authors' responses did not help me better understand the work.
>
> Authors have used some words/terms that I understand their literal meaning, but I don't understand how they are reflected or achieved in this paper. This was also confirmed by other reviewers. I won't repeat the words/terms already pointed out by Reviewers jL87 and judw, but will use authors' responses as examples.
>
> **"1. Evolving world", "2. A new engine proposed to facilitate the evolving world" and their descriptions:** Those are understandable to me. However, they are concepts or a vision. I believe that many researchers, including myself, agree with the vision and would like to see more works towards this direction. But, to achieve this long-term goal, there will be some specific tasks, achievements, attempts, etc. I would like to know, in this particular paper, what was exactly achieved. Like Reviewer KRQt also concerned about "Its codebase will become more and more along with the world’s evolution." Still, I'm not convinced by authors' responses to this comment.
>
> **"3. Training data generation" and its description:** Similar to above, I understand the importance of generating data from LLMs or other generators and agree that novelty and interestingness are important. However, how did the authors measure novelty and interestingness in this work? I also understand that it is hard to measure novelty and interestingness, but how the authors have done exactly in this paper? How does the tag of interestingness work? I searched for "interesting" and found the sentence "Second, code that fails to pass the interestingness threshold will also be discarded" which makes me more confused. what is a "interestingness threshold"? How to set it? Does it change subject to players?
>
> **"4. Potential useful role data" and its description:** I understand the data will be useful, and I checked the asset in the zip, but I failed to identity them as "a large amount of data of Pokemon roles".
>
>
> The core messages I want to share are the followings.
>
> * A good work should be well explained and understandable by readers. Pointing out the writing and clarity issues does not mean that the work is bad. However, it does mean that at least four readers (the reviewers) did not understand the paper. If readers fail to understand the paper and identify the contributions, how can readers evaluate its technical contents or appropriate the work and use your work in other games/virtual environments?
>
> * When a reader finds many things unclear in a paper, it is hard to ask questions or leave comments, because there could be too many. For instance, if I have several questions about *interestingness* as in this response, one may have many more questions about other parts. It is certainly impossible to write a question/comment about every paragraph to understand the paper. Also, one may don't know where to start. This may lead to short review.
>
> * Rebuttal offers a unique chance to clarify some confused contents and answer questions. As the authors have seen, reviewers participated the rebuttal. If reviewers don't want to better understand the paper, reviewers won't spend time on rebuttal. With this idea in mind, authors are suggested to focus on the questions/comments, and behave professionally. Also, please don't push reviewers.
>
> Authors shall check the Code of Conduct at https://iclr.cc/Conferences/2024/CodeOfConduct#:~:text=The%20open%20exchange%20of%20ideas,inherent%20worth%20of%20every%20person.

---

> ### Author Response · Authors · 2024-11-16
>
> Thank you for your peaceful suggestions and willingness to engage in the discussion.
>
> 1. **Evolving world", 2. A new engine proposed to facilitate the evolving world**
>
> Thank you for your reply. We are trying to make our contribution clearer. In this paper, **1.** we propose a scalable engine capable of generating new code based on user behavior. This is our solution for the vision we mentioned. In our initial attempt, we do not make the engine general for all scenarios. We only implement a specified engine on the Pokemon world. **2.** We generate interesting data to enhance the performance of the engine (i.e. mainly out-of-domain performance of code generation). We also share all these data on the web. **3.** Evaluating such a scalable engine is a new task. To do this, we craft two test sets (one is manually crafted) and use two metrics (Exe and Acc, though this is not new). We also design some experiments to access the engine's performance comprehensively.
>
> Besides, we open-source the complete code of our project. We believe there are some researchers would like our work and may build upon our project. We are also committed to make our engine more general, no longer restricted to a single game/virtual world.
>
> 3. **Training data generation and its description**
>
> We explain interestingness vector in Sec. 5.2. It is a one-dimensional vector, which each bit represents one interesting tag (0 or 1). We calculate this magnitude (i.e. sum of all bits) and compare this value with a preset threshold to evaluate the interestingness of a sample.
>
> 4. **Potential useful role data**
>
> Yes. They are not large.

---

### Official Review · Reviewer_KRQt · 2024-11-05

**Soundness:** 1
**Presentation:** 1
**Contribution:** 2
**Rating:** 3
**Confidence:** 4

**Summary:**

In this paper, authors propose a Delta-Engine generates executable code and embed it into the base engine using Large Language Models.

**Strengths:**

In this paper, the authors tried to make the game engine (a backbone of the virtual world) changing over time. It's interesting to update the backbone engine using Large Language Models.

**Weaknesses:**

This paper tries to tackle challenging problems to update game engines (usually static components of the virtual world) over time. However, the solution description on the problem is not clearly defined in the manuscripts.

In the Abstract, authors stated that "existing virtual worlds are strictly defined by the back-end engine and cannot be changed by user's behavior." It's an interesting statement however, it's still questionable that the engine needs to be changed by user's behavior. It's a radical change to the world and there could be a solution to reflect the user's behavior's outcome to the world without changing the game engine itself.

In the Abstract, there are terms that make it difficult for readers to focus on the contribution. For example, they're "scalability to user-generated content," "dual aspects of algorithm and data," "neural proxy," and "novel and interesting data." It's relatively new to be difficult to grasp the concepts from the first reading. It's recommended to improve the summary to be readable for the audience.

In the Introduction, authors argue the necessity of "evolving nature." What's the definition of the evolving in this paper to be used? I recommend authors to provide more explanation why back-bone engines needs to be "evolving" instead of other alternative solutions (evolving objects instead of changing the world itself) traditionally approached in many game-related articles. Also, it's good to add some evidence on "Such dynamics is unpredictable and beyond the reach of existing systems."

Authors need to improve their manuscripts by avoiding unclear definitions of words or terminologies. For example, they're "Its codebase will become more and more along with the world's evolution." "God mode," "Biodiversity," "Imagination," and "Tags of Interest" so on.

In Chapter 3, the Delta-Engine description needs to be improved. For example, the Base engine part includes "only walking ability" "learns to run and even fly." It seems that the engine is limited to the sample scenario. It's desirable to provide a general introduction of the methodology. In the incremental prediction part, please explain what is the input, and what is the value? In the retrieval, what is the sparse version?

In conclusion, this paper's weak point is unclear description of their ideas with ill-defined justification of research goals.

**Questions:**

* Could you generalize the Delta-engine to other problems instead of the Pokemon?
* What is the critical benefit of using Delta-engine instead of traditional game engine (unity, or unreal or custom engines)?
* Could you differentiate your system with an LLM-based story generator? What's the main difference between your work and other narrative generators?
* When you apply Procedural Contents Generation, it assumes the world is not changing over time (e.g., basic rules of the game, or physical property is fixed). How can we modify the current PCG in the context of your changing game engines?

---

> ### Author Response · Authors · 2024-11-14
>
> Thank you for your valuable review.
>
> **W1. Whether the engine needs to be changed to reflect the user's behavior**
>
> Our answer is absolutely yes. One of our main idea is that **every player can be different in the virtual world**, as shown in the film "Ready Player One". The fact is the player base is very large and their choices and largely outweigh the boundary of a static engine. **If the engine is static**, even if it is vey large, **all possibilities will be quickly exhausted by the players**.
>
> **W2. Some new words make it difficult to grasp the contribution of our work**
>
> Thank you for your feedback. We are willing to clarify them in the new revision. On the other hand, we would like to emphasize that our work explores an entirely new scenario and **covers multiple aspects of contributions**, including the problem definition, inference algorithm, and data generation. We hope this broader context can provide clarity and better understanding of the scope of our work.
>
> **W3. Explain the evolving**
>
> Evolving in this paper can be simply understood as the ability of an object in the virtual world to grow new or unknown features. The necessity to evolve, please refer to W1. Thank you.
>
> We are willing to provide the existing evidence for the claim "Such dynamics is unpredictable and beyond the reach of existing systems." Thank you for your suggestion.
>
>
> **W4. Definitions of some words**
>
> "Its codebase will become more and more" -> The code of the engine becomes larger.
>
> "God mode": One can instruct the engine as we wish.
>
> "Biodiversity," "Imagination" are just what the words originally mean.
>
> **"Tags of Interest" is one techniques we propose to evaluate the interestingness in Section 5.2.**
>
>
> **W5.1. Delta-Engine description**
>
> First, learning to run and fly from walking serve as a concrete and toy example we use to illustrate the idea of the delta-engine. **It does not mean that the delta-engine can only be used to this scenario.** Instead, it conveys the general idea of the growing process of the nature.
>
> **W5.2. Input/output for incremental prediction**
>
> For incremental prediction, the input is the user-instruction and the incremental value refers to the new code added to the engine.
>
> The sparse version is detailed in line 153 to line 155. In brief, it is the skeleton overview of the engine's code. For example:
>
> ```python
> PokemonBase
> - get_power
> - get_stats
> - move_1
> ...
> ```
>
> **Q1. Generalize the Delta-engine to other problems**
>
> It is very challenging because different games/virtual worlds are greatly different in their settings, physical laws. This will be the point of our future work. The attempt in our paper is to build a specified delta-engine.
>
> **Q2. Benefit of using Delta-engine**
>
> Thank you for your question. We need to clarify that delta-engine is not a traditional game engine (UE, Unity), which is used for game-development. Instead, delta-engine is an embedded module in the game system to offer new gaming experiences. Its feature is automatically generating new code based on the base engine, which is different from traditional game engines.
>
> **Q3. Difference to narrative generator**
>
> It is unclear what the narrative generators exactly are. Are they used for generating instant plots or stories to push the game development [1] [2]? If they are driven by the user's behavior, then the concept may resemble that of the delta-engine. However, **the technical aspects between the narrative generator and delta-engine is quite different.** The latter focuses the scalability of the engine, e.g. generates the executable code on the base engine, rather than texts. Of course, the texts generated by the narrative generator should also be projected to the action space if it is a game, to my knowledge. To conclude, our work is different from narrative generators and we are willing to discuss this part in the related work. Thank you.
>
> **Q4. How to apply PCG**
>
> Yes, the evolving process should follow the basic rules like physics. This is the potential setting in our work. All these properties are defined in the base engine and will not change. For example, the beating the Fire-type Pokemon can help the learning of a Fire-type move. This is the basic law in our game. Following these unchanged law, we can develop algorithms to generate the procedural content.
>
> Please note that PCG is not our focus in this work. We only focus the delta-engines themselves.
>
> [1] NarrativePlay: Interactive Narrative Understanding
>
> [2] From Role-Play to Drama-Interaction: An LLM Solution

---

### Author Response · Authors · 2024-11-15

As authors, we hope reviewers can join the discussion and provide feedback for us whether our responses address the concerns or not. We are open to making further clarification.

**I acknowledge that there are clarity issues within our submission but we will clarify them as we can. However, it is also likely that most of the reviews offer an unfair rating based on an incomplete understanding of our paper**, which is frustrating for us and ideally should not occur in the ICLR review process.

---

### Author Response · Authors · 2024-11-18

We appreciate the reviewers' work. We update the paper in the rebuttal revision following the ideas in the reviews. Of course, it cannot be the ticket for the acceptance. We try to make our paper better.

1. We greatly modify the abstract and introduction. We find our previous topic too large. In this revision, we choose a smaller one.

2. Update some parts of the related work as some reviewers suggest.

---

### Meta-Review · Area_Chair_zo2X · 2024-12-16

**Metareview:**

The authors should attempt to take on board constructive feedback and improve the paper for future submission. At this point it is below the bar for ICLR. It is unfortunate that the discussion phase was not always a pleasant exchange, but in this case it does not impact the decision since the paper is below the bar for ICLR due to the lack of clarity in the paper as noted by all reviewers.

**Additional Comments On Reviewer Discussion:**

The discussion was unsavory largely due to rudeness from the authors.

---

### Decision · Program_Chairs · 2025-01-22

Reject